# OsMDH12: A Peroxisomal Malate Dehydrogenase Regulating Tiller Number and Salt Tolerance in Rice

**DOI:** 10.3390/plants12203558

**Published:** 2023-10-13

**Authors:** Yuheng Shi, Jiahui Feng, Liping Wang, Yanchen Liu, Dujun He, Yangyang Sun, Yuehua Luo, Cheng Jin, Yuanyuan Zhang

**Affiliations:** 1School of Breeding and Multiplication (Sanya Institute of Breeding and Multiplication), Hainan University, Sanya 572025, China; shiyuheng@hainanu.edu.cn (Y.S.); fjh45797301@163.com (J.F.); 21210901000041@hainanu.edu.cn (L.W.); liuyanchen@hainanu.edu.cn (Y.L.); hdjjjjaptx4869@163.com (D.H.); sunyyhn@hainanu.edu.cn (Y.S.); jincheng@hainanu.edu.cn (C.J.); 2School of Tropical Agriculture and Forestry, Hainan University, Haikou 570228, China; lyhhk@163.com

**Keywords:** rice, malate dehydrogenase, tiller number, salt tolerance

## Abstract

Salinity is an important environmental factor influencing crop growth and yield. Malate dehydrogenase (MDH) catalyses the reversible conversion of oxaloacetate (OAA) to malate. While many MDHs have been identified in various plants, the biochemical function of MDH in rice remains uncharacterised, and its role in growth and salt stress response is largely unexplored. In this study, the biochemical function of *OsMDH12* was determined, revealing its involvement in regulating tiller number and salt tolerance in rice. *OsMDH12* localises in the peroxisome and is expressed across various organs. In vitro analysis confirmed that *OsMDH12* converts OAA to malate. Seedlings of *OsMDH12*-overexpressing (*OE*) plants had shorter shoot lengths and lower fresh weights than wild-type (WT) plants, while *osmdh12* mutants displayed the opposite. At maturity, *OsMDH12*-*OE* plants had fewer tillers than WT, whereas *osmdh12* mutants had more, suggesting *OsMDH12*’s role in tiller number regulation. Moreover, *OsMDH12*-*OE* plants were sensitive to salt stress, but *osmdh12* mutants showed enhanced salt tolerance. The Na^+^/K^+^ content ratio increased in *OsMDH12*-*OE* plants and decreased in *osmdh12* mutants, suggesting that *OsMDH12* might negatively affect salt tolerance through influencing the Na^+^/K^+^ balance. These findings hint at *OsMDH12*’s potential as a genetic tool to enhance rice growth and salt tolerance.

## 1. Introduction

Rice (*Oryza sativa*) is one of the most important food crops worldwide, with over half the global population relying on it as a staple. Tillering is a key agronomic trait that influences grain yield and is affected by both genetic and physiological factors. Therefore, identifying key genes for tiller development is crucial for breeding high-yield crops.

Tiller formation in rice consists of two processes, the initiation of axillary buds on each leaf axil and their further outgrowth [1,2,3]. During the vegetative growth period, axillary buds persist in forming tillers and spikelets to enhance grain yield [4,5]. However, later axillary buds usually go dormant under the regulation of various hormones [5], which is one of the mechanisms through which rice prevents excessive tillering [6]. Previous studies have shown that auxin and cytokinin play a crucial role in regulating the growth of axillary buds in plants [7,8]. Recently, strigolactones (SLs) have been identified as a unique class of terpenoid lactone phytohormones that control a variety of aspects of plant growth and development, including inhibiting the outgrowth of axillary buds [9]. Over the past decade, many genes involved in SL biosynthesis or the SL signalling pathways have been identified, including *DWARF3* (*D3*) [10], *D10* [11], *D14* [12], *D17* [13], *D27* [14], *D53* [15], *HIGH TILLERING DWARF1* (*HTD1*) [16], *MORE AXILLARY GROWTH* (*OsMAX1a*, *OsMAX1b*, *OsMAX1c, OsMAX2d* and *OsMAX1e*) [17], *OsMADS57* [18], *TEOSINTE BRANCHED 1* (*OsTB1*) [19], *IDEAL PLANT ARCHITECTURE 1* (*IPA1*) [20], *OsSHI1* [7], and *CIRCADIAN CLOCK ASSOCIATED1* (*OsCCA1*) [21]. In addition, numerous genes involved in other hormone biosynthesis or the signalling pathway have been identified that control the number of rice tillers [3], including *LAX PANICLE1* (*LAX1*) [22], *LAX2* [23], *MONOCULM1* (*MOC1*) [24], *MOC3* [25], *NITROGEN-MEDIATED TILLER GROWTH RESPONSE 5* (*NGR5*) [26], *Hd3a* [27], *OsDRM2* [28], *Cytokinin oxidase/dehydrogenase* 2 (*CKX2*) [29], *OsGA2s* [30] OsWRKY94 [19], *TIF1* [2], and *Tiller Number 1* (*TN1*) [31]. Currently, cloned genes controlling tiller development in rice mainly regulate the initiation or growth of axillary buds. The discovery of these pivotal genes offers important genetic resources and strategies for improving rice yield traits.

Salinity is an important global environmental factor that limits plant growth and crop productivity. To respond to salt stress, plants have evolved several mechanisms to combine endogenous developmental cues with exogenous salinity stress signals to balance growth and stress responses optimally. Ion toxicity and osmotic stress are major factors in the inhibition of plant growth caused by salt stress [32]. To mitigate the damage caused by osmotic stress under salt stress, plants promote the accumulation of protective metabolites in the cytoplasm [33]. These metabolites can function as compatible osmolytes that do not interfere with plant metabolism, including abscisic acid, flavonoids, proline, sugar, glycine betaine, polyamines, and α-amino nitrogen [34,35]. Under salt stress, these osmolytes play a crucial role in osmoregulation by decreasing cellular osmotic potential and stabilizing proteins and cellular structures [35,36]. The metabolomic profile of various plants (*Arabidopsis*, rice, and lotus) shows that the balance between amino acids and organic acids is a conserved response under salt stress [34,37]. Malate is one of the organic acids that affects plant growth, fruit acidity, and nutrient quality and plays an important role in response to salt stress [38,39,40,41]. Malate is primarily produced in the cytosol and transported to the vacuole for storage as malic acid [42], an intermediate metabolite of the citric and glyoxylate cycles [43]. Many studies have shown that salt stress leads to the accumulation of malic acid in various plants, including *Arabidopsis* [43], Pear [44], apple [45], grape [46,47], rice [48], and tomato [49]. When plants are exposed to salinity stress, malic acid accumulates and is transported into mitochondria, where it feeds into the tricarboxylic acid cycle to promote mitochondrial ATP production and maintain respiratory flux [38,44,50]. Malate dehydrogenase (MDH) is a class of oxidoreductase enzymes that utilise NAD or NADP(H) as cofactors to catalyse the reversible reaction between malic acid and OAA [51]. This reaction is vital for cellular metabolic processes such as the tricarboxylic acid cycle and the malate–aspartate shuttle in both plants and mammals [52]. Various MDHs have been identified in numerous plant species, including *Arabidopsis thaliana* [43], apple [53], Chinese Fir (*Cunninghamia lanceolata*) [54], cotton [55], maize [56], melon [57], poplar (*Populus trichocarpa*) [58], tobacco [59], tomato [60], soybean [61], Stylosanthes (*Stylosanthes guianensis*) [62], and rice [48]. Some have been found to be essential for several anabolic and catabolic processes, converting OAA to malic acid. For instance, *ZmMDH4* [63], *GhmMDH1* [64], and *MdcyMDH1* [40] are known to catalyse the conversion of OAA to malate in maize, cotton, and apples. While the biochemical function of MDH is established, its biological function in plants is not yet fully understood.

In rice, twelve MDH (*OsMDH1*–*OsMDH12*) members are present. *OsMDH1* is a plastid-localised protein that negatively regulates salt tolerance through affecting vitamin B6 content [65]. *OsMDH10*/*FLO16* encodes a NAD-dependent cytosolic malate dehydrogenase. Knockout of *OsMDH10*/*FLO16* notably reduces the starch content in grains, whereas its overexpression significantly increases grain weight, suggesting that *OsMDH10* is vital for starch biosynthesis and seed development [66]. Recent studies indicate that natural variations in the promoter region of *OsMDH8* may be linked to salt tolerance during the rice seedling stage [48]. Most *MDH* family genes are significantly induced by salt stress [48], implying a potential role in rice’s salt stress response. However, the role of the majority of OsMDH genes in plant development and in responding to salt stress remains largely unexplored.

In this study, we functionally characterised *OsMDH12*, which is localised in the peroxisome. We found that *OsMDH12* serves as a negative regulator of both tiller number and salt tolerance in rice.

## 2. Results

### 2.1. Phylogenetic Tree, Expression Profiles, and Subcellular Localization of OsMDH12

The full-length 1071 bp cDNA sequence of *OsMDH12* (Os12g0632700) encodes a malate dehydrogenase that is predicted to be 357 amino acids long, according to the Rice Genome Annotation Project (http://rice.plantbiology.msu.edu (accessed on 1 May 2021)).

A phylogenetic tree was constructed using the amino acid sequences of OsMDH12 and malate dehydrogenases from other plants, such as Arabidopsis, maize, and tomato. OsMDH12 displays high amino acid sequence similarity to AtpMDH1, AtpMDH2, ZmMDH12, ZmMDH13, SlMDH2, SlMDH3, and OsMDH3 (Figure 1A).

To assess the function of *OsMDH12*, we first analysed its expression pattern in various organs. Quantitative real-time PCR (qRT-PCR) revealed that *OsMDH12* is expressed in all tissues examined, including roots, stems, leaves, and spikes (Figure 1B). Subsequently, we explored the expression of *OsMDH12* under abiotic stress conditions. The transcription level of OsMDH12 in rice leaves and roots was detected after salt and cold treatment. The results indicated that the transcript level of *OsMDH12* was strongly induced by both salt and cold stress (Figure 1C,D).

To ascertain the subcellular localization of OsMDH12, the coding sequence of the full-length *OsMDH12* was fused to Green Fluorescent Protein (GFP) under the control of the CaMV35S promoter. Colocalization with the peroxisome protein AtpMDH1 [67] in the leaf epidermal cells of *Nicotiana benthamiana* confirmed a peroxisomal location for OsMDH12 (Figure 1E).

### 2.2. Enzyme Analysis of OsMDH12

Malate dehydrogenase (MDH) has been shown to catalyse the reversible reaction between malic acid and OAA, using NAD or NADP(H) as cofactors (Figure 2A). To ascertain the biochemical function of OsMDH12, we conducted enzymatic assays in vitro using recombinant OsMDH12 proteins. Our results demonstrated that OsMDH12 displayed high activity when OAA was used as the donor (Figure 2B). Conversely, no activity was detected when malate served as the donor (Figure 2B), suggesting that OsMDH12 catalyses the conversion of OAA to malate.

### 2.3. Molecular Characterization of MDH12-OE and osmdh12 Knockout Mutant Plants

To investigate the function of *OsMDH12* in rice, we generated *OsMDH12*-overexpressing (*OsMDH12*-*OE*1 and *OsMDH12*-*OE*2) (Figure 3A) and *osmdh12* CRISPR (*osmdh12*-*1* and *osmdh12*-*2*) lines (Figure 3B)in the japonica cultivar Zhonghua11 (ZH11) background. *osmdh12-1* carried a one-base insertion (T) in the target site, and *osmdh12-1* carried a two-base deletion (TC) in the target site, which all truncated the *OsMDH12* open reading frame (Figure 3A). At the seedling stage, growth in the *OsMDH12*-*OE* plants was inhibited, showing significantly decreased shoot lengths and fresh weights (Figure 3C–E). In contrast, the *osmdh12* mutants displayed a significant increase in shoot lengths and fresh weights(Figure 3C–E). Additionally, malate content was reduced in the *osmdh12* mutant lines compared to the WT (Figure 3F). These results suggested that *MDH12* negatively regulates growth but positively regulates malic acid content in rice.

### 2.4. MDH12 Negatively Regulates Tiller Numbers in Rice

To further explore the physiological functions of *OsMDH12*, we observed the phenotypes of *OsMDH12*-*OE* and *osmdh12* mutant plants at the mature stage. When grown in a paddy field, no significant difference in plant height was observed between the *osmdh12* mutant and the WT (Figure 4A,B). However, *OsMDH12*-*OE* plants displayed fewer tillers compared to WT plants (Figure 4C). Conversely, *osmdh12* plants had a greater number of tillers, indicating that *OsMDH12* negatively regulates tiller numbers in rice (Figure 4C). We then searched a publicly available genetic co-expression database (CREP, http://crep.ncpgr.cn/ (accessed on 1 June 2022)) to identify genes co-expressed with *OsMDH12* that are involved in regulating tiller number in rice. *OsMDH12* exhibited a high correlation coefficient with three genes (*PAY1*, *TN1*, and *OsTOM2*) implicated in tiller number [31,68,69] (Figure 4D, Appendix A). To investigate whether the expression levels of *PAY1*, *TN1*, and *OsTOM2* were affected in *osmdh12* mutant plants, we performed qRT-PCR analysis. The results revealed that the expression of *PAY1* and *TN1* in the shoots was not significantly different between *OsMDH12*-*OE* and *osmdh12* mutant plants compared to WT (Figure 4E,F). However, the expression of *OsTOM2* was repressed in the shoots of *OsMDH12*-*OE* plants but was upregulated in the roots of the *osmdh12* mutant (Figure 4G). These findings suggest that *OsMDH12* regulates tiller number through affecting the expression of *OsTOM2* in rice.

### 2.5. OsMDH12 Negatively Regulates Salt Tolerance in Rice

To explore the role of *OsMDH12* in salt tolerance, we induced salinity stress in *OsMDH12*-*OE* and *osmdh12* mutant plants. Under normal growth conditions, no noticeable phenotypic differences were observed between *OsMDH12* transgenic and WT plants (Figure 5A). However, when subjected to 150 mM NaCl treatment (Figure 5B,C), *OsMDH12*-*OE* plants exhibited salt-sensitive phenotypes, characterised by lower survival rates (Figure 5D) and increased ROS accumulation (Figure 5E). Conversely, *osmdh12* mutants displayed enhanced salt stress tolerance. We then analysed the Na^+^/K^+^ content ratio in both *OsMDH12* transgenic and WT plants. The results revealed that the Na^+^/K^+^ content ratio significantly increased in *OsMDH12*-*OE* plants but decreased in *osmdh12* mutant plants (Figure 5F). These findings suggest that *OsMDH12* negatively regulates salt tolerance through affecting the Na^+^/K^+^ ratio in rice.

## 3. Discussion

In this study, we characterised the function of *OsMDH12*, a member of the rice MDH gene family, and identified its negative regulatory role in controlling tiller number and salt tolerance. Malate dehydrogenase catalyses the reversible conversion of malate and oxaloacetate (OAA) using NADH or NADPH as coenzymes. NADP-dependent MDH isozymes are primarily localised in chloroplasts, while NAD-dependent MDH isozymes are found in other organelles, including the cytoplasm, mitochondria, glycosomes, peroxisomes, and chloroplasts [43]. Our data demonstrate that OsMDH12 is an NADP-dependent MDH isoform which is localised to the peroxisome and catalyses the conversion of OAA to malate in rice (Figure 1 and Figure 2).

Members of the MDH family are crucial for root growth [57], seed germination [66] and maturation [70], and embryo development [43]. In *Arabidopsis*, the suppression of *NAD^+^-dependent plastidial MDH* (*pdnad-mdh*) leads to embryonic lethality, while reduced expression of *pdnad-mdh* results in dwarfism, diminished chlorophyll levels, lowered photosynthetic rates, reduced daytime carbohydrate levels, and chloroplast disorganization [51,68]. In rice, *OsMDH10* has been implicated in starch biosynthesis and seed development [40]. OsMDH12 shares high homology with Arabidopsis AtpMDH1 and AtpMDH2, suggesting functional similarities (Figure 1). In Arabidopsis, the *mmdh1*/*mmdh2* double mutants exhibit significantly reduced growth and photosynthesis rates compared to the WT [69,70]. Similar phenotypes were observed in *osmdh12* mutants during the seedling stage, including decreased plant height and fresh weight (Figure 3). These findings suggest that the functions of *MDH* genes are conserved in both rice and Arabidopsis. However, unlike previous studies on *MDH* genes in plants [43,57,66,71], we found that *OsMDH12* plays a pivotal role in regulating rice tiller number.

Tiller number, a crucial yield component, is a target for genetic improvement in crops [72]. Elements, both macronutrients and micronutrients, significantly influence plant growth and the degree of tillering in rice [68,73]. For example, iron (Fe) limitation hampers rice growth, reduces height, and restricts tiller numbers. *OsTOM2*, part of the major facilitator superfamily (MFS), is a 2’-deoxymugineic acid (DMA) transporter that mediates DMA secretion from roots to the rhizosphere and is involved in Fe mobilization via DMA secretion into the vascular bundles for phloem/xylem metal loading [74]. Inhibition of *OsTOM2* expression markedly reduces tiller number, dry weight, and yield in rice, implying that *OsTOM2* is essential for optimal growth [68]. We observed similar phenotypes in *OsMDH12*-overexpressing plants, which displayed fewer tillers and reduced dry weights (Figure 4). Moreover, the expression of *OsTOM2* was significantly decreased in *OsMDH12*-overexpressing plants, while it increased in *OsMDH12* mutant plants, suggesting that *OsMDH12* may regulate tiller number via *OsTOM2* expression. The role of *OsMDH12* in metal mobilization warrants further investigation.

Several studies have demonstrated that MDH genes play significant roles in plant responses to various abiotic stresses [40,54,57], particularly in salt stress resistance [48,65]. For instance, the overexpression of plastid maize NADP-malate dehydrogenase (*ZmNADP-MDH*) in Arabidopsis enhances its salt stress tolerance [70]. In apples, overexpressing cytosolic NAD-malate dehydrogenase (*MdcyMDH*) results in increased salt tolerance and decreased ROS levels [63]. Recent research has shown that *OsMDH1*, a plastid NAD-dependent dehydrogenase, negatively regulates salt stress in rice [64]. Further analysis indicated that *OsMDH1* modulates the response to salt stress through affecting vitamin B6 levels [64]. Despite the differing subcellular localizations of OsMDH1 and OsMDH12, the phenotypes of transgenic plants overexpressing either gene were similar (Figure 5). Lines overexpressing *OsMDH1* or *OsMDH12* are sensitive to salt stress, while lines with a knockout of *OsMDH1* or *OsMDH12* show tolerance to salt stress. In contrast to OsMDH1, our findings suggest that OsMDH12-mediated salt tolerance is closely linked to an imbalance in the K^+^/Na^+^ ratio. Whether *OsMDH12* is involved in the regulation of vitamin B6 synthesis remains an open question for further study.

In conclusion, we have uncovered the biological function of *OsMDH12*, a member of the malate dehydrogenase family. Our results show that *OsMDH12* encodes a peroxisomal malate dehydrogenase that converts OAA to malate. Moreover, we demonstrate that *OsMDH12* plays an essential role in regulating both plant growth and the response to salt stress.

## 4. Materials and Methods

### 4.1. Phylogenetic Analysis

The OsMDH protein sequences were extracted from the TIGR database (http://rice.uga.edu/ (accessed on 1 May 2021)). The MDH protein sequences of Arabidopsis (AtMDH), tomato (SlMDH), and maize (ZmMDH) were obtained from the National Center for Biotechnology Information (NCBI, Bethesda, MD, USA, https://www.ncbi.nlm.nih.gov/ (accessed on 1 May 2021)). These sequences were aligned using MEGA 6 (Temple University, Philadelphia, PA, USA) software, and a neighbour-joining (NJ) tree was generated with 1000 bootstrap replicates; all other parameters were set to their default values.

### 4.2. Plant Materials and Stress Treatments

All plant materials used in this study were either ZH11 or derived from this cultivar. The *osmdh12* mutants were generated using the CRISPR-Cas9 system, while *OsMDH12*-*OE* plants were produced under the control of the maize ubiquitin promoter in ZH11. The full-length coding sequence of *OsMDH12* was amplified from Nipponbare (NIP) cultivar and cloned into the pJC034 vector. For physiological analyses, seedlings were cultivated in paddy fields in Lingshui, Hainan Province, China. For salt treatments, seeds were germinated for 3 days at 37 °C and then incubated in culture solution at 28 °C for 7 days. Following this, the seedlings were transferred to a 150 mmol/L NaCl solution for an additional 7 days. Seeds were then allowed to recover in a nutrient solution for 7 days [75]. Shoot and root growth seedings were estimated as surviving seedings. Thirty plants of ZH11 and *OsMDH12* transgenic lines for each experiment were used for subsequent statistical calculations. For cold treatment, the seedlings were exposed to 6 °C [76]. Leaves were sampled at 1 h, 3 h, 6 h, 9 h, 12 h, and 24 h intervals.

### 4.3. Gene Transcription Analysis

qRT-PCR was conducted to assess gene expression following salt or cold treatments. Leaves from various treatment groups were harvested at different time points post-treatment. All samples were processed using the RNA isolator Total RNA Extraction Reagent (Vazyme, Nanjing, China), and first-strand cDNA was synthesised using EasyScript One-step gDNA Removal and cDNA Synthesis SuperMix (TRANSGEN, Beijing, China), as per the manufacturer’s instructions. Primers for gene specificity were designed using Primer 5.0 and are listed in Appendix A. The internal reference used was the Ubiquitin5 gene. qRT-PCR was carried out on an ABI 7500 Real-Time PCR system (Applied Biosystems, Waltham, MA, USA), utilizing a SYBR Premix Ex Taq Kit in accordance with the manufacturer’s guidelines. All reactions were conducted in a minimum of three biological replicates.

### 4.4. Subcellular Localization

The full-length coding sequence of *OsMDH12* was cloned into the pH7GWFS2.0 vector to generate 35S:EGFP-*OsMDH12*. Similarly, AtpMDH1, a peroxisome marker, was cloned into the same vector to produce 35S:RFP-AtpMDH1 [60]. Both 35S:EGFP-*OsMDH12* and 35S:RFP-AtpMDH1 were transiently expressed in *N. benthamiana* leaves with agrobacterium EHA105 and imaged using a laser confocal microscope (LSM980; Zeiss, Jena, Germany) after 2 days of incubation at 22 °C.

### 4.5. OsMDH Activity Assays

The full-length coding sequence of *OsMDH12* was cloned into the pDEST15 vector using the Gateway recombination reaction (Invitrogen, Waltham, MA, USA). Expression constructs were introduced into BL21 (DE3) cells, and the expression of the fusion proteins was induced with 0.1 mM IPTG when the bacterial culture reached an OD600 of 0.5. After incubation at 16 °C for 14 h, cells were harvested via centrifugation and resuspended in 50 mL of Lysis Buffer containing Tris-HCl (50 mM), NaCl (50 mM), 5% glycerol, and PMSF (0.1 mM). Cells were disrupted using a pressure crusher, and the soluble protein fractions were subsequently centrifuged [77]. Purification was carried out and monitored using a GST resin filter column for GST-tagged proteins. OsMDH12 activity was assessed through the spectrophotometric measurement of NADH consumption at 340 nm. The reaction was conducted in a 250 mM HEPES (pH 8.0) buffer containing MgCl_2_ (2 mM), NAD^+^/NADH (0.25 mM), and either OAA or malate (2.5 mM). Purified OsMDH12-GST or GST (5 μg) was added to a 200 μL reaction volume. All reactions were conducted in a minimum of three replicates.

### 4.6. Measurement of Malate

The liquid chromatography–tandem mass spectrometry (LC-MS/MS) system (QTRAP 6500, AB SCIEX, Toronto, ON, Canada) was used to determine the content of malate as previously described [73]. All samples were rushed using a high-throughput tissue grinder (MM400, Retsch, Haan, Germany ) after freeze-drying. The dry powder was extracted with 70% aqueous methanol containing lidocaine as the internal standard (0.1 mg/L) overnight at 4 °C. Following centrifugation at 1000× *g* for 5 min, the lipids were absorbed and filtered using a 0.22 μm pore size nylon syringe filter (SCAA-104, ANPEL, Shanghai, China) before the LC-MS analysis. For LC-MS analysis, a C18 column (Shim-pack GISS, SHIMADZU, Kyoto, Japan, 2.1 mm × 100 mm, 1.9 μm) was equipped. A 2 μL sample was eluted by mobile phase at 0.3 mL/min flow rate at 40 °C. Three biological replicates were measured for each transgenic line.

### 4.7. NBT Staining

For the NBT assay, small pieces of the second leaf from 14-day-old seedlings were vacuum-filtered in 0.1% (*w*/*v*) NBT for 30 min and then incubated for 10 h under ambient conditions. After draining the solution, 95% ethanol was added to decolourise the sample at 80 °C until the solution became colourless.

### 4.8. Determination of Sodium and Potassium Concentrations

The rice was harvested seven days after the initiation of salt treatment and was rinsed three times with distilled water. Samples were digested in nitric acid in a microwave (Multivave Pro 41, Anton Paaranton Paar, Graz, Austria) for 1 h, as previously described [78]. Sodium and potassium ion concentrations were measured using ICP-MS (ICP-MS 7800, Agilent, Santa Clara, CA, USA) after dilution in deionised water. Three biological replicates were measured for each transgenic line.

### 4.9. Statistical Analysis

Plant phenotype, the content of malate, gene expression, survival rate, and Na^+^/K^+^ ratio were analysed via one-way or two-way ANOVA using Microsoft Excel software (Office_Professional_Plus_2016) and corrected with Student’s *t*-test at a significance level of 0.05.

### 4.10. Accession Numbers

The accession numbers of genes in this article are as follows: OsMDH12 (Os12g0632700), PAY1 (Os08g0407200), TN1 (Os01g0610300), and OsTOM2 (Os11g0135000). Sequence data from this article can be found in the Rice Genome An-notation Project website (http://rice.plantbiology.msu.edu/ (accessed on 1 May 2021)) and NCBI (https://www.ncbi.nlm.nih.gov/ (accessed on 1 May 2021)).

## Figures and Tables

**Figure 1 plants-12-03558-f001:**
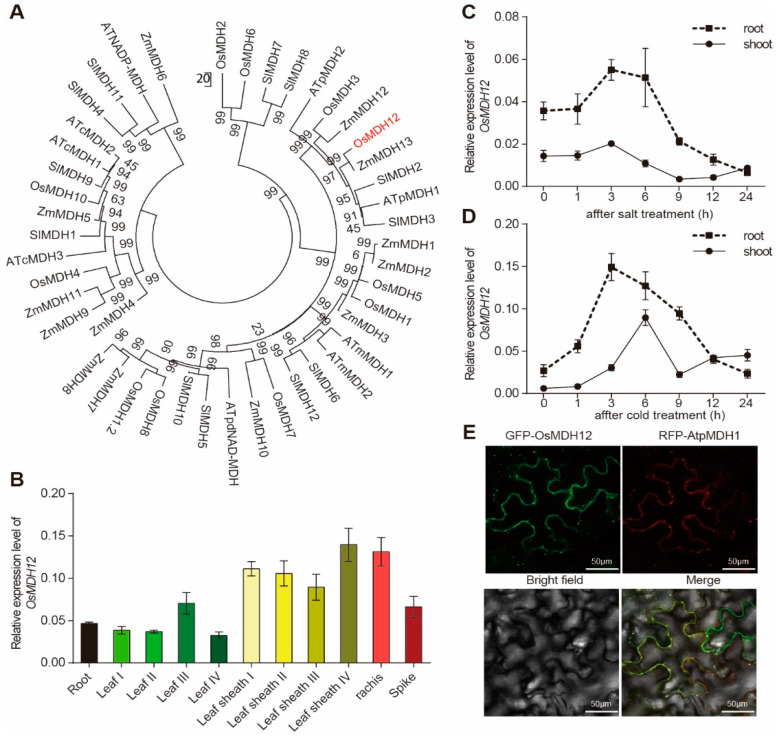
Expression pattern of *OsMDH12*. (**A**) Phylogenetic relationship of MDH proteins in rice, *Arabidopsis*, and tomato. Bootstrap values from 1000 trials are indicated. (**B**) Expression levels of *OsMDH12* in different rice organs. Samples were taken from Zhonghua11 grown in a paddy field. (**C**,**D**) Time-course expression of *OsMDH12* under salt (**C**) and cold (**D**) treatment. Standard deviations are based on three biological replicates. (**E**) Subcellular localization of OsMDH12 protein in leaves of *N. benthamiana*. 35S:GFP-OsMDH12 and 35S:RFP:AtpMDH1 were co-transformed into the leaves of *Nicotiana benthamiana*. AtpMDH1 serves as a peroxisome marker. The scale is set at 50 μm.

**Figure 2 plants-12-03558-f002:**
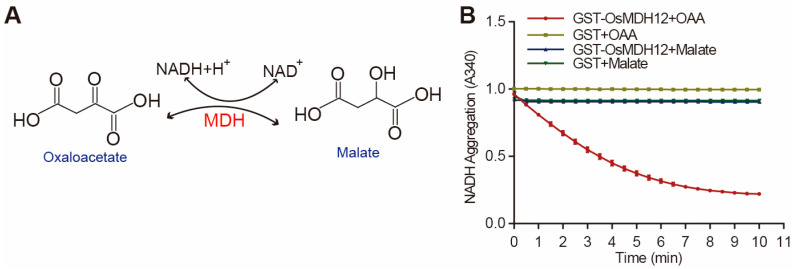
OsMDH12 catalyses OAA to malate. (**A**) Schematic diagram of the enzyme-catalysed reaction by Bend OsMDH12. (**B**) Assays of OsMDH12 activity. The assay was conducted in the presence of GST-OsMDH12, GST, NADH, and OAA. Data are expressed as the decrease in OD at 340 nm over time.

**Figure 3 plants-12-03558-f003:**
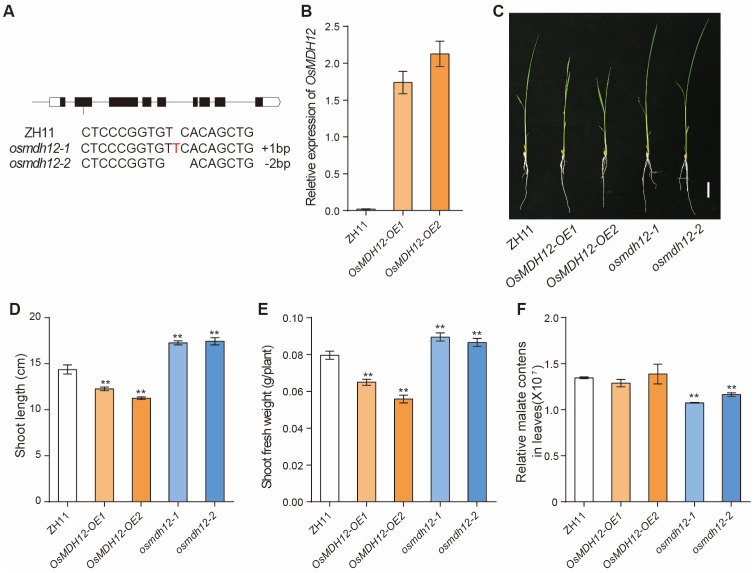
Appearance and malate contents of *OsMDH12* transgenic lines at the seedling stage. (**A**) The *osmdh12* knockout mutants were generated using CRISPR gene editing technology. (**B**) Expression level of *OsMDH12* in two independent overexpressing lines. (**C**) Phenotype comparison of ZH11, *OsMDH12*-*OE* lines, and *osmdh12* mutants (two different CRISPR lines) at the seedling stage. The scale is set at 2 cm. (**D**,**E**) Plant height (**D**) and fresh weight (**E**) in ZH11, *OsMDH12*-*OE*, and *osmdh12* mutants. (**F**) Malate content in ZH11, *OsMDH12*-*OE*, and *osmdh12* mutants. Values are presented as mean ± SD (n = 10); ** *p* < 0.01 (*t*-test).

**Figure 4 plants-12-03558-f004:**
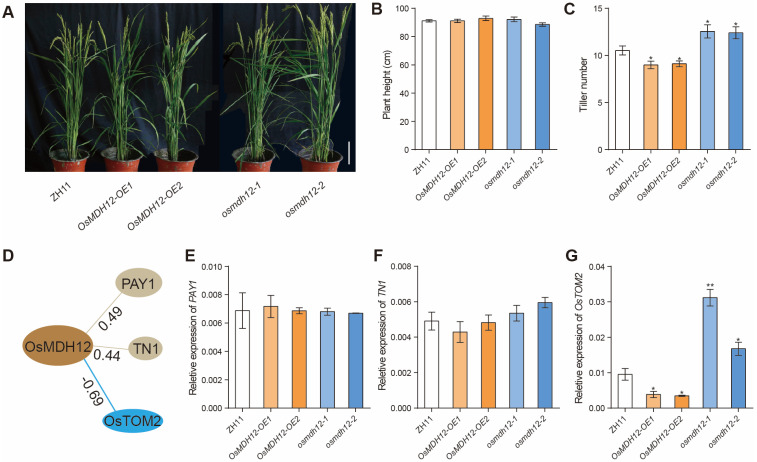
*OsMDH12* regulates tiller numbers in rice. (**A**) Phenotypic comparison between WT and *OsMDH12* transgenic plants at the maturity stage. The scale is set at 10 cm. (**B**,**C**) Plant height and tiller number in ZH11, *OsMDH12*-*OE*, and *osmdh12* mutant. (**D**) Genes co-expressed with *OsMDH12* related to tillering. (**E**–**G**) Expression levels of *PAY1* (**E**), *TN1* (**F**), and *OsTOM2* (**G**) in ZH11, *OsMDH12*-*OE*, and *osmdh12* mutant. Values are presented as the mean ± SD (n = 10); * *p* < 0.05, ** *p* < 0.01 (*t*-test).

**Figure 5 plants-12-03558-f005:**
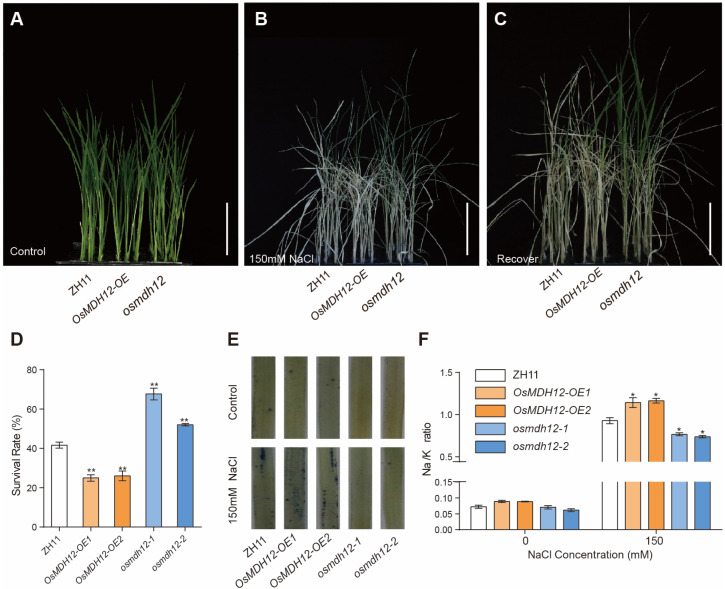
Performance of *OsMDH12* transgenic plants under salt stress. (**A**–**C**) Two-week-old rice seedlings from ZH11, *OsMDH12*-*OE*, and *osmdh12* plants (**A**) were treated with 150 mM NaCl for seven days (**B**) and then allowed to recover for three days (**C**). The scale is set at 5 cm. (**D**) Survival rate under salt stress. (**E**) Phenotype revealed by NBT staining after 24 h of salt stress. (**F**) Na^+^/K^+^ ratio in ZH11, *OsMDH12*-*OE*, and *osmdh12* mutants. Values represent the mean ± SD (n = 3); * *p* < 0.05, ** *p* < 0.01 (*t*-test).

## Data Availability

All data are available upon reasonable request.

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
