# Peer review of "OsMDH12: A Peroxisomal Malate Dehydrogenase Regulating Tiller Number and Salt Tolerance in Rice"

_plants, 2023, doi:10.3390/plants12203558_

Round 1

Reviewer 1 Report

Article OsMDH12: A Peroxisomal Malate Dehydrogenase Regulating Tiller Number and Salt Tolerance in Rice

by authors Yuheng Shi, Jiahui Feng, Liping Wang, Yanchen Liu, Dujun He, Yangyang Sun, Yuehua Luo, Cheng Jin, Yuanyuan Zhang provides data from a detailed study of the role of malate dehydrogenase in salt tolerance in rice.

The manuscript contains the necessary sections and is formatted in accordance with the rules.

The material is presented logically and illustrated.

Captions under the figures contain complete and sufficient information.

Minor drawbacks: the bar is missing in Figures 4 and 5.

There is no statistics section in the materials and methods, which does not allow us to understand the legality of using the selected type of histograms.

The introduction is too short and does not allow the idea of the study to be appreciated.

It is recommended to increase the size of the pictures.

I would like to expand the physiological explanation of the results obtained.

Reviewer 2 Report

The research is interesting and provide good information on the regulation of tiller and stress tolerance through OsMDH. I have few minor concern, which is listed below.

1. Figure 3C phenotype is not very clear in their pixel.

2. What is the impact of MDH12 on photosynthesis (as it catalyze the reaction between malic acid and OAA), in overexpressed and mutant lines?

3. Have you investigated MDH12's role in other abiotic stress in overexpressed and mutant lines?

English is fine to me.
